# Early Metabolomic Profiling as a Predictor of Renal Function Six Months After Kidney Transplantation

**DOI:** 10.3390/biomedicines12112424

**Published:** 2024-10-22

**Authors:** Iris Viejo-Boyano, Marta Isabel Roca-Marugán, María Peris-Fernández, Julián Luis Amengual, Ángel Balaguer-Timor, Marta Moreno-Espinosa, María Felipe-Barrera, Pablo González-Calero, Jordi Espí-Reig, Ana Ventura-Galiano, Diego Rodríguez-Ortega, María Ramos-Cebrián, Isabel Beneyto-Castelló, Julio Hernández-Jaras

**Affiliations:** 1Nephrology Department, University and Polytechnic La Fe Hospital, 46026 Valencia, Spain; maria_peris@iislafe.es (M.P.-F.); moreno.marta2206@gmail.com (M.M.-E.); mafeba1993@hotmail.com (M.F.-B.); gonzalezcalero_pab@gva.es (P.G.-C.); espi_jor@gva.es (J.E.-R.); ventura_ana@gva.es (A.V.-G.); rodriguez_die@gva.es (D.R.-O.); ramos_marceb@gva.es (M.R.-C.); beneyto_isacas@gva.es (I.B.-C.); hernandez_jul@gva.es (J.H.-J.); 2Nephrology Unit, Health Research Institute Hospital La Fe, 46026 Valencia, Spain; 3Analytical Unit Platform, Health Research Institute Hospital La Fe, 46026 Valencia, Spain; marta_roca@iislafe.es; 4Big Data AI and Biostatistics Platform, Health Research Institute Hospital La Fe, 46026 Valencia, Spain; julian_amengual@iislafe.es (J.L.A.); angel_balaguer@iislafe.es (Á.B.-T.)

**Keywords:** kidney transplant, metabolomics, biomarkers

## Abstract

Background: Kidney transplantation is the therapy of choice for patients with advanced chronic kidney disease; however, predicting graft outcomes remains a significant challenge. Early identification of reliable biomarkers could enhance post-transplant management and improve long-term outcomes. This study aimed to identify metabolomic biomarkers within the first week after kidney transplantation that predict renal function at six months. Methods: We conducted a prospective study involving 50 adult patients who received deceased donor kidney transplants. Plasma samples collected one week after transplant were analyzed using liquid chromatography–mass spectrometry in a semi-targeted metabolomic approach. A Partial Least Squares-Discriminant Analysis (PLS-DA) model identified metabolites associated with serum creatinine > 1.5 mg/dL at six months. Metabolites were selected based on a Variable Importance in Projection (VIP) score > 1.5, which was used to optimize model performance. Results: The PLS-DA model demonstrated strong predictive performance with an area under the curve (AUC) of 0.958. The metabolites negatively associated with serum creatinine > 1.5 mg/dL were 3-methylindole, guaiacol, histidine, 3-indolepropionic acid, and α-lipoic acid. Conversely, the metabolites positively associated with worse kidney graft outcomes included homocarnosine, 5-methylcytosine, xanthosine, choline, phenylalanine, kynurenic acid, and L-kynurenine. Conclusions: Early metabolomic profiling after transplantation shows promise in predicting renal function. Identifying metabolites with antioxidant and anti-inflammatory properties, as well as those that are harmful and could be targeted therapeutically, underscores their potential clinical significance. The link between several metabolites and the tryptophan pathway suggests that further specific evaluation of this pathway is warranted. These biomarkers can enhance patient management and graft survival.

## 1. Introduction

Chronic kidney disease (CKD) represents a significant global health problem, affecting approximately 12% of the population and leading to high rates of morbidity and mortality that continue to rise [1]. Kidney transplantation is the therapy of choice for patients with advanced CKD. However, predicting graft outcomes remains a major challenge in post-transplant management, highlighting the urgent need for new biomarkers, particularly those that can be identified early in the post-transplant period to predict long-term outcomes.

Metabolomics offers a promising approach for biomarker discovery, including untargeted metabolomics, which provides a comprehensive analysis of all detectable metabolites and has been used in urine samples to monitor and predict graft function [2]; targeted metabolomics, which focuses on the precise quantification of specific metabolites and has been applied to serum samples to assess metabolic profile changes during the first six months after transplant [3]; and semi-targeted metabolomics, which combines elements of both approaches to provide a focused yet comprehensive analysis [4], though it has not yet been evaluated for predicting graft evolution in kidney transplantation.

Our objective is to identify reliable biomarkers within the first week following kidney transplantation using a semi-targeted approach with an in-house compound library. These biomarkers should be associated with serum creatinine levels exceeding 1.5 mg/dL at six months, a threshold linked to adverse long-term outcomes in previous studies [5]. By enabling rapid diagnosis and early intervention, we aim to improve kidney transplant (KT) outcomes and enhance patient prognosis.

## 2. Materials and Methods

### 2.1. Study Design and Participants

This prospective, observational, single-center study included 50 adult patients with stage 5 CKD requiring renal replacement therapy who received a deceased donor KT, with follow-up extending over six months after transplant. Inclusion criteria encompassed patients over 18 years old selected to receive a compatible deceased donor KT. Exclusion criteria included patients with advanced CKD not requiring renal replacement therapy, those on immunosuppressive drugs prior to the transplant, and those receiving combined transplants.

### 2.2. Data Collection

Clinical variables of both donors and recipients, along with baseline analytical data, were collected before transplant (T0). Clinical outcomes, analytical variables, and metabolomic samples were subsequently collected at two time points: T1 (one week after transplant) and T2 (six months after transplant). For T1, samples were specifically collected on either day 7 or day 8 after transplant to ensure that, for patients requiring hemodialysis, samples were obtained before the dialysis session to avoid any potential alterations caused by the hemodialysis process.

Blood samples for metabolomic analysis were collected and processed by centrifugation. The resulting plasma was stored at −80 °C until study completion.

### 2.3. Metabolomic Analysis

For the metabolomic analysis, 190 μL of cold methanol was added to 10 μL of plasma to precipitate proteins and extract compounds. The mixture was vortexed for 30 s and incubated at −20 °C for 30 min. Subsequently, the samples underwent double centrifugation at 9000 rpm for 10 min at 4 °C. From the resulting supernatant, 90 μL was transferred to a 96-well plate for liquid chromatography–mass spectrometry and 10 μL of an internal standard mix solution containing phenilanaline-d5, caffeine-d9, reserpine, and leucine enkephalin at 20 μM was added. Quality control (QC) samples were prepared by pooling 5 μL from each serum sample. A blank sample was prepared by replacing serum with ultrapure water to identify potential artifacts from tubes, reagents, and other materials.

Samples were analyzed using an Ultra-Performance Liquid Chromatography (UPLC) system coupled to a Q-Exactive Plus spectrometer. Chromatographic separation was performed with an XBridge BEH Amide column (150 × 2 mm, 2.5 μm particle size; Waters). The autosampler was set to 4 °C, and a flow rate of 95 μL/min was used with water, 20 mM ammonium acetate, and 20 mM ammonium hydroxide at pH 9.4 as Mobile Phase A and acetonitrile as Mobile Phase B. To minimize intra-batch variability and enhance reproducibility, samples were injected in random order, with a QC sample analyzed in every six plasma samples. Blank analyses were conducted at the end of the sequence.

Electrospray ionization was employed in both positive and negative modes, using full mass acquisition, with a resolving power of 140,000 and three mass ranges: 70–300 Da, 300–560 Da, and 350–1200. Data were acquired in centroid mode. Raw data were converted to mzXML format using MSConvert and then processed in EI-MAVEN software (v0.11.0) with an in-house polar compound library to generate a peak table containing *m*/*z* values, retention times, and intensities of detected metabolites.

Data quality (reproducibility, stability) was assessed using the internal standard’s stability and the QC’s coefficients of variation (CVs). Molecular features with CVs > 30% were excluded from the data matrix. Additionally, LOESS normalization was applied to eliminate intra-batch variability due to technical differences. Data from positive and negative ionization modes were merged and filtered for statistical analysis.

### 2.4. Statistical Analysis

Initially, descriptive statistical analyses were performed on the baseline variables of both donors and recipients, as well as on basic laboratory values and clinical outcomes. Subsequently, a statistical analysis was conducted on the metabolomic data obtained at time point T1, using serum creatinine levels greater than 1.5 mg/dL at six months as the outcome variable. The metabolomic dataset included 226 molecules from 48 patients (initially 50 patients; one patient was excluded due to early graft loss and another due to a sample error at T1). The categorical response variable (creatinine levels at six months after transplant) had two levels: 26 patients with creatinine ≤ 1.5 mg/dL and 22 patients with creatinine > 1.5 mg/dL.

For the metabolomic analysis, a Partial Least Squares-Discriminant Analysis (PLS-DA) model was initially constructed using all available metabolites. The model utilized seven principal components, and the optimal number of latent variables was determined through Leave-One-Out Cross-Validation (LOOCV) to minimize classification error. A Variable Importance in Projection (VIP) score threshold of 1.5 was applied, identifying 22 polar metabolic variables as significant contributors to the initial model.

A second PLS-DA model was then constructed using only the 22 selected variables obtained from the first model. This final model used two principal components, with the number of latent variables again optimized using LOOCV to ensure minimal classification error. Model validation was performed through permutation testing with 1000 iterations, confirming that the R^2^ coefficient obtained with the original response variable was greater than that obtained with the permuted response variable in 95% of cases. This provided an empirical *p*-value for model significance, enhancing the robustness of our findings.

Performance metrics of the final model were evaluated based on LOOCV predictions, including the receiver operating characteristic (ROC) curve, area under the curve (AUC), accuracy, sensitivity, and specificity. These metrics provided a comprehensive assessment of model performance. Additionally, coefficients and 95% confidence intervals (CIs) for each variable were calculated using the Jack-Knifing resampling technique. Only significant metabolomic variables were highlighted as the most relevant for clinical interpretation, emphasizing their potential significance in relation to the outcome variable.

## 3. Results

The study cohort included 50 KT recipients with a mean age of 53.98 years, predominantly male (70%) and Caucasian (94%). Cardiovascular risk factors were common: 82% had hypertension, 70% had dyslipidemia, and 28% were classified as obese (Body Mass Index, (BMI) > 30). The primary etiologies of CKD were hereditary/familial causes (26%) and unclassified causes (24%).

Regarding renal replacement therapy (RRT) prior to transplantation, 74% of recipients were on hemodialysis and 26% on peritoneal dialysis, with a mean RRT duration of 3.44 years. Most recipients had residual diuresis of less than 500 mL (64%). Sensitization was observed in 18% of recipients, with 6% being highly sensitized (calculated Panel Reactive Antibody (cPRA) > 98%). The mean Estimated Post-Transplant Survival (EPTS) score was 41.96%.

Donors had a mean age of 50.58 years, with 70% being female. Most donors were from donation after brain death (DBD) (58%), and 32% were expanded criteria donors (ECDs). The mean Kidney Donor Profile Index (KDPI) was 52.66, and the mean cold ischemia time was 17.30 h. Full demographic and clinical details are summarized in Table 1.

The semi-targeted metabolomic approach identified 226 polar compounds. The final PLS-DA model results are presented in Figure 1 and Table 2. The scores plot (Figure 1A) illustrates the projection of the observations (patients) onto the two components, which together account for 26.6% of the total X-variance (Component 1: 17.1%; Component 2: 9.5%). The two groups of subjects exhibit clear clustering based on creatinine levels (creatinine >1.5 mg/dL and creatinine ≤1.5 mg/dL), although some partial overlap is observed. Figure 1B provides the corresponding loading plot, illustrating how each metabolite contributes to the separation between the two groups. The metabolites included in the final PLS-DA model are detailed in Table 2, along with their chemical formulas, spectrometric properties, and statistical parameters, including standardized coefficients, confidence intervals, and *p*-values obtained through Jack-Knifing resampling. Metabolites with a significant *p*-value are highlighted in red in Figure 1B.

We identified twelve statistically significant metabolites. Five of them have negative standardized coefficients, indicating their association with creatinine ≤ 1.5 mg/dL: 3-methylindole (skatole), guaiacol, histidine, 3-indolepropionic acid, and α-lipoic acid. The remaining seven have positive standardized coefficients, indicating their association with creatinine > 1.5 mg/dL: homocarnosine, 5-methylcytosine, xanthosine, choline, phenylalanine, kynurenic acid, and L-kynurenine.

The model performance metrics after cross-validation are as follows: the model achieved an AUC of 0.958, with an overall accuracy of 0.875, a sensitivity of 0.8636, and a specificity of 0.8846. The model correctly identified 19 true positives and 23 true negatives, while misclassifying 3 false positives and 3 false negatives. The ROC curve, shown in Figure 1C, further illustrates the model’s performance, with the curve approaching the top-left corner.

## 4. Discussion

The evolution of kidney graft function in the initial months after transplant correlates with long-term outcomes. Delayed graft function (DGF) and acute rejection are among the most significant, though not the only, determinants of renal disease progression [6]. Monitoring involves serial assessments of renal function, urine sediment analysis, proteinuria, antibody measurements, and microbiological tests, with renal biopsy being the most accurate but invasive method [7]. This has led to a growing interest in identifying non-invasive biomarkers that can predict KT outcomes earlier and more reliably.

Omics technologies, particularly metabolomics, have emerged as promising tools for biomarker discovery. In KT, omics studies have primarily focused on diagnosing acute rejection [8,9,10]. However, few studies have examined other events such DGF [11] or overall progression [3]. A notable study by Ihsan Yozgat et al. [2] used untargeted metabolomics of urine samples to develop a PLS-DA model, correlating a panel of marker candidates with serum creatinine levels above 1.5 mg/dl one year after transplant, achieving an AUC of 0.9 for the one-week post-transplant model. This study highlighted the potential of a metabolite panel for improved discrimination but noted the challenge of collecting urine samples from DGF patients.

Our study used semi-targeted plasma metabolomics at one week after transplant, leveraging a precise in-house compound library. Using a UPLC system coupled to a Q-Exactive Plus spectrometer and applying a PLS-DA model for statistical analysis, we achieved excellent performance metrics, with an AUC of 0.958 following cross-validation.

We identified five statistically significant metabolites with negative standardized coefficients, indicating their association with better graft function and suggesting a protective role. These metabolites include 3-methylindole (skatole), guaiacol, histidine, 3-indolepropionic acid, and α-lipoic acid.

3-Methylindole, or skatole, is a metabolite derived from tryptophan metabolism via the indole pyruvate pathway, facilitated by gut microbiota through the enzyme tryptophan dehydrogenase (TrpD) [12] (Figure 2). Sin-Hyoung Hong et al. demonstrated its protective effects by reducing endoplasmic reticulum stress, oxidative stress, lipogenesis, caspase activity, and apoptosis in hepatocytes [13]. Although its direct effect on the kidney have not been established, a review suggests potential benefits on CKD through maintaining intestinal homeostasis, given the link between dysbiosis and CKD [14]. However, high concentrations of skatole are associated with colorectal cancer and pneumotoxicity [15], indicating the need for further investigation.

Guaiacol, or 2-methoxyphenol, is a phenolic compound known for its antioxidant properties, as it can donate electrons and hydrogen atoms to neutralize free radicals and prevent lipid peroxidation [16]. Its antioxidant activity also includes the reversible inhibition of myeloperoxidase, an enzyme that promotes oxidative stress and inflammation, leading to cardiovascular disease complications [17,18]. Although the relationship between guaiacol and renal disease is not well described, its ability to reduce oxidative stress and inflammation may help protect kidney graft function, as suggested by our study.

Histidine is an amino acid with an imidazole side chain that provides proton buffering, metal ion chelation, and antioxidant properties. It scavenges reactive oxygen species (ROS) and reactive nitrogen species (RNS) and sequesters advanced glycation end products (AGEs) and advanced lipoxidation end products (ALEs) [19]. Lower levels of histidine have been observed in CKD patients [20], and it is used to treat CKD-related anemia [21]. Its protective properties have led to its inclusion in organ preservation solutions such as Custodiol^®^, which is used in our patients.

3-Indolepropionic acid (IPA) is a tryptophan metabolite produced by gut microbiota through the enzyme tryptophanase (TNA) [12] (Figure 2). IPA exhibits potent antioxidant activities, scavenging hydroxyl radicals and improving gut barrier function [22]. It downregulates fibrosis-related genes and inflammatory factors, providing significant renal protection. Additionally, IPA competes with toxic substances like indoxyl sulfate for organic anion transporters, reducing their accumulation in renal tubules and mitigating renal injury [23]. Finally, IPA is recognized as a potential biomarker for protection against CKD [24], and our study associates it with the prevention of adverse renal graft outcomes.

α-Lipoic acid is a dithiol compound that acts as a cofactor for several enzymes and serves as an energy modulator. It exchanges thiol groups with other thiol-containing metabolites, making it a redox modulator and a potent antioxidant by reducing ROS and RNS, chelating metals, and regenerating antioxidants like vitamins C and E. It also has anti-inflammatory properties, targeting NF-κB and decreasing inflammatory cytokines [25]. Its protective effects are documented in various renal diseases, including diabetic nephropathy, sepsis-associated acute kidney failure, nephrotoxicity-induced renal failure, and ischemia-reperfusion (I/R) injury [26]. In I/R injury, α-lipoic acid combats oxidative stress, downregulates Na-K-ATPase and nitric oxide synthase, mitigates neutrophil infiltration, inhibits inflammation, and suppresses endothelin-1 upregulation [27]. It is used as a therapeutic strategy in various solid organ transplants [25] and could serve as a biomarker for protection against adverse renal graft outcomes.

Conversely, our study identified seven statistically significant metabolites with positive standardized coefficients, indicating their association with worse kidney graft outcomes. These metabolites include homocarnosine, 5-methylcytosine, xanthosine, choline, phenylalanine, kynurenic acid, and L-kynurenine.

Homocarnosine, a dipeptide of gamma-aminobutyric acid and histidine, has been associated with various biological and pathological brain events. Its role in kidney diseases is less studied; however, elevated levels have been observed in diabetic kidney disease, compared with individuals with type 2 diabetes mellitus without nephropathy [28], and in renal tissue following ischemia [29], which may be related to the I/R injury observed in our KT recipients.

5-Methylcytosine (5-mC) is a methylated form of cytosine where a methyl group is attached to the carbon at position 5. DNA methylation has been extensively studied in the context of KT and involves the addition of methyl groups to cytosines in CpG dinucleotides, which cluster in CpG islands, leading to the transcriptional silencing of affected genes [30]. Although methylation is relatively stable, it can be reversed by TET enzymes, which convert 5-mC to 5-hydroxymethylcytosine in an oxygen-dependent manner. Hypoxia during KT reduces TET activity, leading to the accumulation of 5-mC and the downregulation of genes involved in suppressing kidney fibrosis and injury [31]. This hypermethylation can predict chronic allograft injury [31], as observed in our study, where elevated 5-mC levels predicted worse outcomes at six months.

Xanthosine is a purine nucleoside identified as a prognostic marker for CKD in several studies, often used alongside other metabolites, such as kynurenic acid, kynurenine, and choline, which was also identified in our study, to estimate CKD progression [32,33,34]. The mechanism by which xanthosine increases CKD risk and its relationship with KT remains unclear. However, purine metabolism dysregulation in CKD, nucleotide breakdown [35] in I/R injury, and the association of other purine metabolites like hypoxanthine as a marker of ischemia [36] could be related. An in vitro study showed that hypoxia significantly decreased purine nucleotide phosphorylase activity [37], the enzyme required to convert xanthosine to xanthine, possibly explaining the accumulation of xanthosine in these patients.

Choline, an essential vitamin found in phospholipids, is involved in methylation reactions, such as converting homocysteine to methionine via its oxidation to betaine. It is linked to CKD progression through its conversion to trimethylamine-N-oxide, which promotes renal fibrosis [34,38]. However, its association with KT appears to be more closely related to I/R injury. Choline is a marker of tissue ischemia linked to phospholipid degradation, exhibiting a biphasic release pattern with a peak immediately following global ischemia and a delayed phase during reperfusion [39]. Lindeman J et al. observed choline uptake in kidney graft recipients from living or deceased donors without DGF, while those with DGF showed choline release, partially masked by its conversion to betaine, potentially associated with the role of betaine in methylation processes occurring in these patients [40]. These findings align with our study, as DGF is related to subsequent renal function and the deleterious effects of elevated choline levels on long-term renal function.

Phenylalanine is an essential amino acid that serves as a precursor for various compounds. Its primary metabolic pathway is its conversion to tyrosine, which can then follow multiple metabolic routes, including the synthesis of catecholamines, melanin, and energy metabolism through deamination and conversion to acetoacetate and fumarate, intermediates of the Krebs cycle. The kidney is the major producer of tyrosine in the body, converting phenylalanine to tyrosine via phenylalanine hydroxylation. Consequently, impaired renal function leads to the accumulation of phenylalanine and a deficiency of tyrosine [41], both linked with CKD progression and fibrosis [42] due to their roles in energy metabolism. In KT, I/R injury decreases phenylalanine hydroxylase activity, leading to phenylalanine buildup [43] and poorer graft outcomes.

The kynurenine pathway (KP) is the main route for tryptophan catabolism, essential for energy metabolism and immune modulation (Figure 2). Excessive activation of this pathway is associated with various kidney diseases, including AKI, CKD, KT progression, I/R injury, infections in KT, and graft rejection. Elevation of these metabolites is linked to inflammation, with pro-inflammatory cytokines like IFN-γ upregulating indoleamine 2,3-dioxygenase (IDO), the rate-limiting enzyme of KP, while downregulating enzymes such as kynureninase and quinolinate phosphoribosyltransferase. This imbalance leads to the accumulation of KP intermediates and disrupts de novo NAD+ biosynthesis [44,45]. Elevated KP metabolites, such as kynurenine and kynurenic acid (KYNA), correlate with poor renal outcomes and renal fibrosis [34,46]. Notably, kynurenine had the highest score as a biomarker for CKD in a large population analysis in [47].

In KT, increased tryptophan degradation and elevated kynurenine levels correlate with worse kidney function [48], can predict graft rejection [49], and are associated with infections due to the inflammation present in both conditions, raising questions about their specificity as rejection markers. A study by Dharnidharka et al. [50] found that the serum kynurenine/tryptophan (KYN/TRP) ratio was higher in cases of acute graft rejection. However, peripheral blood CD4-ATP levels were useful in differentiating between rejection and infection, being lower in the infection group. Our study supports the importance of monitoring KP metabolites like kynurenine and KYNA as potential prognostic biomarkers and therapeutic targets in CKD and KT patients.

This study has several limitations. First, the small sample size, due to the high cost and technical complexity of metabolomic studies, limits the ability to perform subgroup analyses, such as distinguishing between recipients of brain-dead versus cardiac-death donors, to avoid loss of statistical power. Second, metabolomic studies are inherently exploratory and hypothesis-generating, providing a foundation for future, more comprehensive research that requires validation in larger cohorts. Third, although the long-term renal graft outcomes have been correlated with renal function at six months in other studies, the short follow-up period in our study does not allow us to confirm these long-term outcomes, making further investigation in this area necessary. Lastly, the use of a semi-targeted approach with an in-house library of compounds, aimed at limiting irrelevant results and improving characterization, may have restricted the discovery of other unknown metabolites.

## 5. Conclusions

In conclusion, our study reveals the important role of biomarkers in predicting KT outcomes. We identified five protective metabolites, which predominantly exhibit antioxidant and anti-inflammatory properties in other studies and could serve as therapeutic targets in KT recipients. Additionally, this study highlights the significant relationship between metabolites from tryptophan metabolism and the prediction of KT evolution, paving the way for further research, therapeutic strategies, and improved and earlier patient management.

## Figures and Tables

**Figure 1 biomedicines-12-02424-f001:**
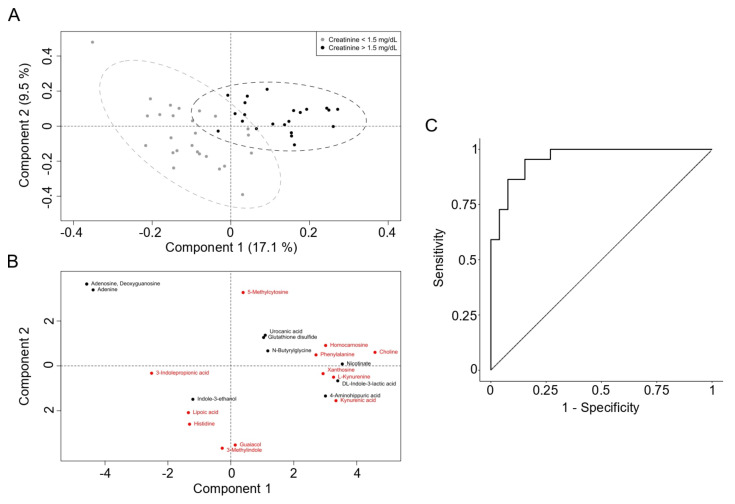
Score plot of PLS-DA based on metabolomics. (**A**). The samples are colored based on the two levels, with grey representing creatinine ≤ 1.5 mg/dL and black representing creatinine > 1.5 mg/dL. The ellipses around each cluster represent the 95% confidence regions, indicating the spread and overlap of each group. (**B**) Loading plot of PLS-DA showing the contribution of the 22 selected metabolites (whose VIP scores were over 1.5 in the first PLS-DA model) to the two categories. Each point represents a metabolite, and its position indicates its influence on the two components. Red points indicate significant *p*-values for the variable (Cr > 1.5 mg/dL). (**C**) Receiver operating characteristic (ROC) curve illustrating the performance of the LOOCV of the PLS-DA model in distinguishing patients with creatinine ≤ 1.5 mg/dL or > 1.5 mg/dL six months after the KT. The diagonal dashed line represents the baseline performance of a random classifier. Dot point shows the optimal cutoff point based on the maximal specificity and sensibility, which corresponds to 0.5.

**Figure 2 biomedicines-12-02424-f002:**
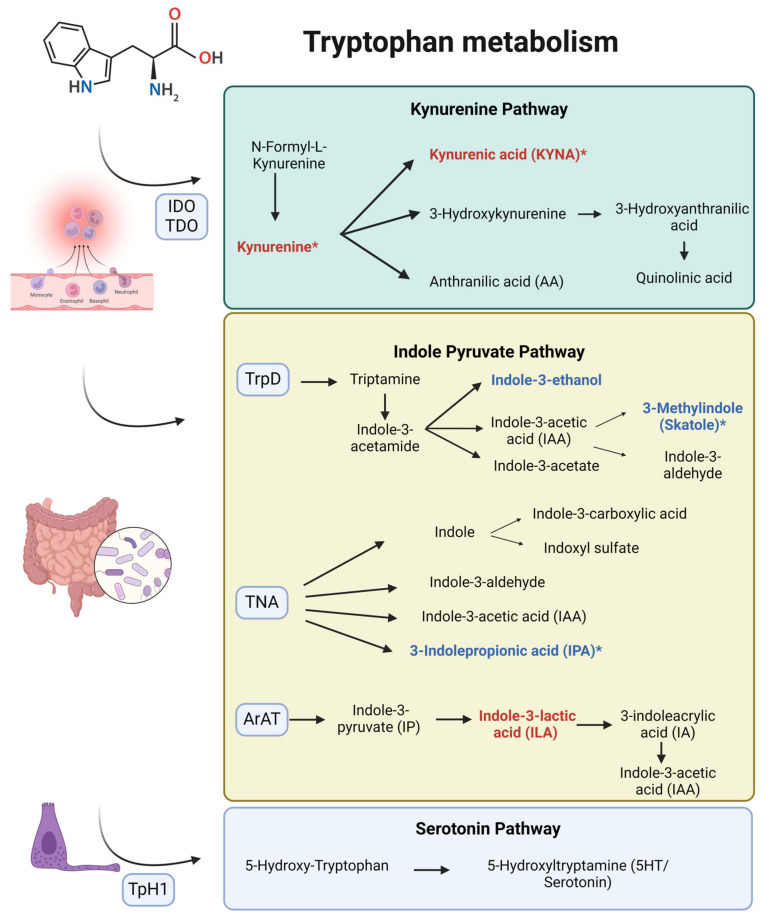
Tryptophan metabolism pathways and associated metabolites with post-transplant renal function. This diagram illustrates the key pathways of tryptophan metabolism: the kynurenine pathway, indole pyruvate pathway, and serotonin pathway. Metabolites marked in red are associated with an increased risk of having a serum creatinine level greater than 1.5 mg/dL at six months after transplant. Metabolites marked in blue are associated with a reduced risk (protective effect) of having a serum creatinine level greater than 1.5 mg/dL at six months after transplant. Metabolites marked with an asterisk (*) denote statistically significant associations. ArAT: aromatic amino acid transaminase; IDO: indolamine-2,3-dioxygenase; TDO: tryptophan 2,3-dioxygenase; TNA: tryptophanase; TpH1: tryptophan hydroxylase 1; TrpD: tryptophan dehydrogenase.

**Table 1 biomedicines-12-02424-t001:** Demographic and clinical characteristics of the sample.

Kidney Transplant Recipient	
Variable	Mean/Frequency
Age (years) (mean ± SD)	53.98 ± 10.94
Gender, *n* (%)	Male: 35 (70%) Female: 15 (30%)
Race, *n* (%)	Caucasian: 47 (94%)
	African American: 3 (6%)
Blood type, *n* (%)	0+: 18 (36%)
	0−: 1 (2%)
	A+: 19 (38%)
	A−: 6 (12%)
	B+: 5 (10%)
	B− 0 (0%).
	AB+: 1 (2%)
	AB−: 0 (0%)
Hypertension, *n* (%)	41 (82%)
Type 2 diabetes, *n* (%)	8 (16%)
Dyslipidemia, *n* (%)	35 (70%)
BMI (mean ± SD)	26.58 (3.89)
Obesity, *n* (%)	Underweight: 0 (0%)
	Normal weight: 18 (36%)
	Overweight: 18 (36%)
	Obesity: 14 (28%)
Hyperuricemia, *n* (%)	17 (34%)
Smoking status, *n* (%)	Non-smoker: 21 (42%)
	Former smoker: 20 (40%)
	Current smoker: 9 (18%)
Physical activity, *n* (%)	Sedentary: 39 (78%)
	Moderately active: 5 (10%)
	Very active: 6 (12%)
Etiology of chronic kidney disease, *n* (%)	Glomerulonephritis: 5 (10%)
	Chronic pyelonephritis/tubulointerstitial: 7 (14%)
	Diabetes mellitus: 6 (12%)
	Hypertension/vascular diseases: 3 (6%)
	Hereditary/familial: 13 (26%)
	Systemic diseases: 4 (8%)
	Unclassified: 12 (24%)
Renal replacement therapy, *n* (%)	Hemodialysis: 37 (74%)
	Peritoneal dialysis: 13 (26%)
Time on dialysis (years) (mean ± SD)	3.44 ± 2.32
Residual diuresis, *n* (%)	<500 mL: 32 (64%)
	500–1000 mL: 7 (14%)
	>1000 mL: 11 (22%)
Heart failure, *n* (%)	2 (4%)
Coronary artery disease, *n* (%)	7 (14%)
Vascular disease, *n* (%)	4 (18%)
Previous transplant, *n* (%)	6 (12%)
Transfusions history, *n* (%)	17 (34%)
Pregnancy history, *n* (%)	12 (24%)
Sensitization, *n* (%)	No: 41 (82%)
	<98% PRAc: 6 (12%)
	>98% PRAc (PATHI): 3 (6%)
EPTS, (mean ± SD)	41.96 ± 26.94
**Kidney Donor**	
**Variable**	**Mean/Frequency**
Age (mean ± SD)	50.58 ± 16.30
Gender, *n* (%)	Male: 15 (30%)
	Female: 35 (70%)
Donor type, *n* (%)	DBD: 29 (58%)
	DCD: 21 (42%)
Hypertension, *n* (%)	14 (28%)
Diabetes mellitus, *n* (%)	9 (18%)
BMI (mean ± SD)	26.29 ± 5.85
Donor AKI, *n* (%)	2 (4%)
Expanded criteria donor (EC), *n* (%)	16 (32%)
KDPI (mean ± SD)	52.66 ± 28.72
**Transplant Process**	
**Variable**	**Mean/Frequency**
Cold ischemia time (mean ± SD)	17.30 ± 4.58
Mismatch 6/6 (mean ± SD)	4.3 ± 1.18
Mismatch 10/10 (mean ± SD)	7.2 ± 1.82
**One-Week Events**	
**Variable**	**Mean/Frequency**
Overdose of calcineurin inhibitor, *n* (%)	30 (60%)
Urinary infection, *n* (%)	11 (22%)
Graft rejection, *n* (%)	3 (6%)
Graft function, *n* (%)	Immediate graft function: 10 (20%)
	Slow graft function: 13 (26%)
	Delayed graft function: 27 (54%)
**Six-Month Post-Transplant Events (Excluding Week 1)**	
**Variable**	**Mean/Frequency**
Graft rejection, *n* (%)	1 (2%)
Urinary infection, *n* (%)	18 (36%)
CMV infection*, n* (%)	3 (6%)
BK infection, *n* (%)	Yes (viremia): 5 (10%)
	Nephropathy: 0 (0%)
MACE	6 (12%)

AKI: Acute Kidney Injury. BMI: Body Mass Index. cPRA: calculated Panel Reactive Antibody. EPTS: Estimated Post-Transplant Survival. KDPI: Kidney Donor Risk Index. MACEs: Major Adverse Cardiovascular Events. PATHI: National Plan for Access to Kidney Transplantation for Highly Sensitized Patients. SD: Standard Deviation.

**Table 2 biomedicines-12-02424-t002:** Compounds used for the final PLS-DA model for predicting renal function (Cr > 1.5 mg/dL).

Compound	Formula	Properties	Standardized Coefficient	*p*-Value	Confidence Interval
**3-Methylindole (Skatole)**	C_9_H_9_N	m/z: 132,080,765 Rt: 3.46 Adduction: [M+H]+	−0.16695299	**0.012**	**−0.29594 to −0.03795**
**Guaiacol**	C_7_H_8_O_2_	m/z: 125,059,898 Rt: 2.811 Adduction: [M+H]+	−0.15992134	**0.040**	**−0.31206 to −0.00777**
**Homocarnosine**	C_10_H_16_N_4_O_3_	m/z: 241,129,028 Rt: 13.951 Adduction: [M+H]+	0.15774800	**0.010**	**0.03892 to 0.27657**
**5-Methylcytosine**	C_5_H_7_N_3_O	m/z: 126,066,383 Rt: 8.141 Adduction: [M+H]+	0.15492272	**0.023**	**0.02254 to 0.28729**
**Histidine**	C_6_H_9_N_3_O_2_	m/z: 156,076,614 Rt: 13.985 Adduction: [M+H]+	−0.14781219	**0.002**	**−0.23791 to −0.05770**
**Xanthosine**	C_10_H_12_N_4_O_6_	m/z: 283,068,512 Rt: 8.486 Adduction: [M-H]-	0.14141625	**0.029**	**0.01564 to 0.26718**
**Choline**	C_5_H_13_NO	m/z: 104,107,384 Rt: 15.016 Adduction: [M+H]+	0.13956783	**0.028**	**0.01596 to 0.26317**
Glutathione disulfide	C_20_H_32_N_6_O_12_S_2_	m/z: 613,161,499 Rt: 13.655 Adduction: [M+H]+	0.12867136	0.065	−0.00778 to 0.26512
**3-Indolepropionic acid**	C_11_H_11_NO_2_	m/z: 188,071,854 Rt: 2.661 Adduction: [M-H]-	−0.12850942	**0.001**	**−0.19942 to −0.05759**
Nicotinate	C_6_H_5_NO_2_	m/z: 124,039,261 Rt: 8.114 Adduction: [M+H]+	0.12825902	0.074	−0.01247 to 0.26899
**α-Lipoic acid**	C_8_H_14_O_2_S_2_	m/z: 207,051,132 Rt: 2.506 Adduction: [M+H]+	−0.12588711	**0.047**	**−0.25017 to −0.00160**
**Phenylalanine**	C_9_H_11_NO_2_	m/z: 166,086,014 Rt: 8.604 Adduction: [M+H]+	0.11587927	**0.041**	**0.00471 to 0.22704**
**Kynurenic acid**	C_10_H_7_NO_3_	m/z: 188,035,202 Rt: 4.198 Adduction: [M-H]-	0.11255399	**0.039**	**0.00591 to 0.21918**
**L-Kynurenine**	C_10_H_12_N_2_O_3_	m/z: 209,091,873 Rt: 8.843 Adduction: [M+H]+	0.11071323	**0.020**	**0.01812 to 0.20330**
Urocanic acid	C_6_H_6_N_2_O_2_	m/z: 139,050,247 Rt: 7.991 Adduction: [M+H]+	0.10565113	0.233	−0.06970 to 0.28100
N-Butyrylglycine	C_6_H_11_NO_3_	m/z: 146,081,177 Rt: 7.097 Adduction: [M+H]+	0.10469998	0.061	−0.00492 to 0.21432
Indole-3-lactic acid	C_11_H_11_NO_3_	m/z: 204,066,437 Rt: 2.615 Adduction: [M-H]-	0.10158428	0.097	−0.01908 to 0.22225
4-Aminohippuric acid	C_9_H_10_N_2_O_3_	m/z: 195,076,187 Rt: 5.703 Adduction: [M+H]+	0.10137408	0.081	−0.01293 to 0.21568
Índole-3-ethanol	C_10_H_11_NO	m/z: 162,091,446 Rt: 2.668 Adduction: [M+H]+	−0.09839586	0.137	−0.22901 to 0.03222
Adenine	C_5_H_5_N_5_	m/z: 134,047,058 Rt: 7.545 Adduction: [M-H]-	−0.03343797	0.339	−0.10300 to 0.03612
Adenosine	C_10_H_13_N_5_O_4_	m/z: 268,104,065 Rt: 7.39 Adduction: [M+H]+	−0.02616427	0.308	−0.07729 to 0.02496
Deoxyguanosine	C_10_H_13_N_5_O_4_	m/z: 268,104,065 Rt: 7.39 Adduction: [M+H]+	−0.02616427	0.308	−0.07729 to 0.02496

The table shows the compounds from the final PLS-DA model with their chemical formulas, spectrometric properties, standardized coefficients, confidence intervals, and *p*-values (Jack-Knifing resampling) for predicting Cr > 1.5 mg/dL. Positive standardized coefficients indicate that increases in the variable are associated with a higher likelihood of Cr > 1.5 mg/dL, while negative coefficients suggest a higher likelihood of Cr ≤ 1.5 mg/dL. Compounds with statistical significance (*p* < 0.05) are highlighted in bold. m/z: mass-to-charge ratio (*m*/*z*). Rt: retention time (in minutes).

## Data Availability

The data supporting the findings of this article are included in the main text of the manuscript. Additional data, such as all metabolomic results, are available upon request.

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
