# Peer review of "Early Metabolomic Profiling as a Predictor of Renal Function Six Months After Kidney Transplantation"

_biomedicines, 2024, doi:10.3390/biomedicines12112424_

Round 1
Reviewer 1 Report
Comments and Suggestions for Authors
The manuscript proposes to uncover metabolimic profiling to address the effectiveness of kidney transplant. This is a huge landmark for transplantation science, since there is still incontable problems associated.
The manuscript is very clear addressing the question, bringing thoughtful insights into scientific community. I would like to see this applied nowadays on clinic and a prospective study in the hospital to verify the numbers of transplant rejection decreasing.
For my point of view, this is the right way to use such powerful tools as metabolomics. It innovates the way research findings and tools could be applied to real life improvements!
The manuscript is well written, has interesting and easy to understand images, as citations look accurate and clear; also, with a good introductory section, the results based on the methodology are spot on and the conclusions raised several interesting points to be investigated in the future (like the main involved metabolism, that may be further investigated to prevent transplant rejection).
I recommend this manuscript to publication.
Author Response
Dear Reviewer,
Thank you very much for your positive feedback on our manuscript titled "Early metabolomic profiling as a predictor of renal function six months after kidney transplantation." We're glad you found the study clear and meaningful, and we appreciate your encouraging words regarding its impact on the field of transplantation.
We fully agree that applying metabolomics in the clinic and conducting prospective studies to track transplant rejection rates is a valuable next step. It’s something we’re very interested in exploring further.
Thank you again for your kind words about the manuscript. We will keep refining the work based on the feedback we've received.
Best regards,
Iris Viejo-Boyano
Reviewer 2 Report
Comments and Suggestions for Authors
The manuscript titled with "Early metabolomic profiling as a predictor of renal function six months after kidney transplantation" demonstrates the potential of early metabolomic profiling to predict long-term kidney graft outcomes. The research is promising and can be highlighted with several key points:
1. The predictive performance is excellent compared with homogeneous researches. A strong predictive ability, with an AUC of 0.95, can be proved for the differentiation of patient based on their six-month seron creatinine levels.
2. Extensive protective as well as harmful metabolites have been detected and identified with LCMS. Those metabolites were associated with better or worse kidney graft outcomes, and provide insights into potential therapeutic targets.
3. The authors indicated the significant relationship between tryptophan metabolism and kidney transplant. It is of interest to people in pharmaceutical and medicinal areas and can be further explored in those areas.
Overall, the study is well-designed and the results are logical and well-established. Although the kidney recipient and donor cohort is not ideal because it is a small group, but it could be understood for only research purpose. Therefore, I would recommend to publish this manuscript as it is.
Author Response
Dear Reviewer,
Thank you for your thoughtful feedback on our manuscript titled "Early metabolomic profiling as a predictor of renal function six months after kidney transplantation." We are pleased to hear that you found the predictive performance, especially the AUC of 0.95, compelling, and that the identification of both protective and harmful metabolites provided valuable insights.
We also appreciate your interest in the relationship between tryptophan metabolism and kidney transplantation. We agree that this area holds great potential for further exploration, particularly in the pharmaceutical and medicinal fields.
Regarding the cohort size, we acknowledge its limitations, but as you mentioned, it was structured for research purposes, and we believe that these findings can serve as a solid foundation for future larger-scale studies.
We are grateful for your recommendation to publish the manuscript and for your encouraging words about the design and results of the study.
Best regards,
Iris Viejo-Boyano
Reviewer 3 Report
Comments and Suggestions for Authors
The authors evaluated renal function six months after kidney transplantation in 50 patients.
The supplementary materials contain valuable information. Resultados_polares_trasplante should be explained more. Table S1 should not be in the supplementary materials but in the text. The authors should explain the results of Table S1 more. In the Conclusion, the authors should include the data from Table S1.
Comments on the Quality of English LanguageTable S1. should be in the text, not the supplementary materials.
Resultados_polares_trasplante should be explained more.
Author Response
Dear Reviewer,
I greatly appreciate your useful comments and suggestions. As you rightly pointed out, Table S1 contains valuable information for the manuscript, so I have now included it in the main text as Table 2. I have also added a footnote to help better understand the content of the table. Additionally, the results in the conclusion were already arranged by the highest standardized coefficients, and as you mentioned, including this table in the main body of the article is indeed helpful (Please see the attachment).
Regarding Resultados_polares_trasplante, these data are not intended for publication in the manuscript but will be available upon request. For this reason, I have also modified the Data Availability Statement at the end of the manuscript to the following:
Data Availability Statement: The data supporting the findings of this article are included in the main text of the manuscript. Additional data, such as all metabolomic results, are available upon request.
This dataset provides the metabolomic data obtained, including identified metabolites, quality control samples, and results for each patient.
Thank you again for your constructive feedback.
Best regards,
Iris Viejo-Boyano
Round 2
Reviewer 3 Report
Comments and Suggestions for Authors
The author moved Table 2. Compounds used for the final PLS-DA model for Predicting Renal Function (Cr >1.5 200
mg/dL) in the text. However, the authors did not provide any explanations in the text. Even There is no Table 2 in the text.
The authors should discuss between Table 2 and Figure 1. Scores plot of PLS-DA based on metabolomics.
Comments on the Quality of English LanguageEnglish is still difficult to understand.
Author Response
Dear Reviewer,
Thank you for your valuable feedback and for the opportunity to improve our manuscript. We have carefully addressed the points you raised:
-
Clarification of Table 2 in the Text: We apologize for the confusion regarding the explanations for Table 2. In the revised manuscript, we have ensured that Table 2 is explicitly referenced and discussed in the text. Following your suggestion, we have included a discussion that elaborates on the connection between Table 2 and Figure 1. This includes a detailed explanation of how the metabolites with significant p-values, highlighted in Table 2, drive the separation of groups observed in Figure 1.
-
Quality of English Language: We acknowledge your comment regarding the readability of the manuscript. In response, we have made substantial revisions to improve the clarity and fluency of the text.
We hope that these revisions adequately address your concerns and contribute to a clearer and more informative presentation of our findings. We are grateful for your suggestions and remain committed to further enhancing the quality of our work.
Thank you again for your time and consideration.
Sincerely,
Iris Viejo-Boyano

Round 3
Reviewer 3 Report
Comments and Suggestions for Authors
no more comments.
Comments on the Quality of English Languageno more comments.